# Validity and Reliability of Inertial Motion Unit-Based Performance Metrics During Wheelchair Racing Propulsion

**DOI:** 10.3390/s25061680

**Published:** 2025-03-08

**Authors:** Raphaël Ouellet, Katia Turcot, Nathalie Séguin, Alexandre Campeau-Lecour, Jason Bouffard

**Affiliations:** 1Département de Kinésiologie, Faculté de Médecine, Université Laval, Québec, QC G1V 0A6, Canada; raphael.ouellet.3@ulaval.ca (R.O.); katia.turcot@kin.ulaval.ca (K.T.); 2Centre Interdisciplinaire de Recherche en Réadaptation et Intégration Sociale (CIRRIS), CIUSSS de la Capitale Nationale, Québec, QC G1M 2S8, Canada; alexandre.campeau-lecours@gmc.ulaval.ca; 3Club d’athlétisme de l’Université Laval, Québec, QC G1V 0A6, Canada; nathalie.seguin.4@ulaval.ca; 4Département de Génie Mécanique, Faculté des Sciences et Génie, Université Laval, Québec, QC G1V 0A6, Canada

**Keywords:** adapted sports, reproducibility of results, biomechanics, accelerometry, inertial sensor

## Abstract

This study aims to evaluate the concurrent validity and test–retest reliability of wheelchair racing performance metrics. Thirteen individuals without disabilities and experience in wheelchair racing were evaluated twice while performing maximal efforts on a racing wheelchair. Three wheelchair athletes were also assessed to compare their performance with novice participants. The wheelchair kinematics was estimated using an inertial motion unit (IMU) positioned on the frame and a light detection and ranging (Lidar) system. The propulsion cycle (PC) duration, acceleration, average speed, speed gains during acceleration, and speed loss during deceleration were estimated for the first PC and stable PCs. The test–retest reliability was generally moderate (0.50 ≤ ICC < 0.75) to good (0.75 ≤ ICC < 0.90), while few metrics showed poor reliability (ICC < 0.50). High to very high correlations were obtained between both systems for 10 out of 11 metrics (0.78–0.99). Wheelchair athletes performed better than novice participants. Our results suggest that integrated accelerometer data could be used to assess wheelchair speed characteristics over a short distance with a known passage time. Such fine-grain analyses using methods usable in the field could allow for data-informed training in novice and elite wheelchair racing athletes.

## 1. Introduction

Considering the recent popularity rise in adaptive sports, notably wheelchair sports, the need for assessment tools to precisely quantify sports performance is becoming apparent [1,2]. Wheelchair propulsion is a complex movement that originates from the upper extremities and trunk segments. When these segments apply force to the wheel’s pushrim, it results in the acceleration of the user–wheelchair system. During wheelchair racing (WR), one of the most practiced wheelchair sports, the variation in speed within each propulsion cycle (PC) is also influenced by the inertial parameters of the user–wheelchair system, which evolve dynamically based on the athlete’s movements [3,4,5]. Therefore, several factors, such as the equipment or propulsion techniques, will greatly influence the performance. Quantifying the speed variations during WR can enable the assessment of the effects equipment adaptations or training strategies with robust performance variables [4]. Hence, the kinematics and spatiotemporal metrics of the PC are considered fundamental to optimize the performance during wheelchair sport [6,7]. Unfortunately, few performance assessment technologies are adapted to quantify these speed variations, mainly because algorithms are often developed for sports involving steps or because the data acquisition frequency is too low [8]. Accordingly, there is a call for technology development in adapted sports, and, more specifically, for wheelchair propulsion [2,9,10,11].

To meet this need, there is an emerging interest in the use of sensors (e.g., global positioning system (GPS), light detection and ranging (Lidar), and inertial motion units (IMUs)) in adaptive sports within the scientific literature [12]. Among those technologies, IMUs have the advantage of offering an analysis of kinematics and spatiotemporal metrics with low interference due to their small weight and size [13,14]. They also offer the spatiotemporal sensibility required for quantifying individual PCs. Because of these characteristics, they were the main assessment tools for wheelchair sports among the technologies reported in the scientific literature [12]. The emergence of studies using the IMU shows promising results. The placement of the IMU on the frame and axle were accurate for monitoring spatiotemporal parameters using contact identification and showed sufficient accuracy for the in-field kinetic assessment of performance [15,16,17]. IMUs were also placed on the spokes of the wheel for evaluating the influence of the trunk angle on the speed and speed variation, using algorithms developed for wheelchair court sports [4].

Despite the growing interest for WR in the adapted sports literature, there is a gap concerning psychometric properties (e.g., validity and reliability) for IMU-based metrics in this sport [12,18]. The validity and test–retest reliability are key concepts to evaluate the quality of the measurement and to allow for its interpretation [11,19,20]. To optimize the effectiveness and the applicability of the technology in WR, these properties need to be measured. This lack of quality measurements is concerning, especially when compared to wheelchair court sports, where extensive IMU-based kinematics metrics show good validity [21,22,23,24,25] and intra-test reliability [21,26,27].

This project aims at developing and evaluating the psychometric properties of spatiotemporal and speed-based metrics using a single IMU positioned on the frame of the wheelchair. It is the first step in developing a feedback system for WR athletes. Therefore, the objective of this study was to evaluate the convergent validity (IMU vs Lidar systems) and test–retest reliability of wheelchair racing performance metrics in able-bodied individuals that are novice in WR. We expected good-to-excellent reliability (intraclass correlation coefficient [ICC] > 0.80) and high correlations (r ≥ 0.70) between the IMU and Lidar systems based on a previously published study [25]. As a secondary objective, we qualitatively compared the performance of novice participants (NPs) and wheelchair racing athletes (WRAs) using the validated metrics. It was expected that the WRAs would perform better than the NPs according to most of the extracted metrics. Prior to the presentation of the main experiment, the following section introduces sensor-based data collection and processing methods to evaluate sprinting performance in various sports.

## 2. State of the Art

Sprinting velocity can be estimated using various types of sensors, each having their own strengths and limitations. As presented previously, multiple measurement quality (e.g., reliability, validity, and spatiotemporal sensitivity) and usability (e.g., low interference and ease of use) factors need to be considered for a proper integration of performance assessment metrics during training. IMUs have the potential to satisfy these criteria, considering their high portability and their ability to provide data at a high sampling frequency both indoors and outdoors. This section summarizes the current use of IMUs and of the isolated sensors composing them (i.e., the gyroscope and accelerometer) in the literature to estimate the linear velocity during various sprinting sports.

During stand-up sprint, multiple approaches can be used to estimate the running velocity based on IMU sensor data [28]. IMUs placed close to the center of mass (e.g., the lumbar spine and pelvis) [29] or on the foot have been used [30]. Whichever sensor location, the initial step consists of calibrating the IMU to express data in the runner frame of reference, isolating the forward acceleration [29]. Then, the forward velocity is computed by integrating the forward acceleration data. Such an integration of accelerometer data leads to an accumulation of errors and a significant drift in the resulting velocity estimate, quickly making it unusable. Several strategies were implemented to correct the data for this drift. These strategies generally consist in correcting the drift by fusing the high-frequency, drifting, IMU-based velocity estimate with a low-frequency, but more stable, velocity measurement. For example, Setuain, Lecumberri [29] used the split time collected with photoelectric cells placed at known distances (between 0.5 m and 20 m) to estimate the drift and to correct the acceleration data using a least-square optimization method. While these authors used discrete measures of the average velocity to correct for the IMU integration drift, others used continuous velocity signals, taken from a GPS sensor, for example, alongside a Kalman or complementary filter [31]. Similar approaches were used in other sports, including paraswimming [32] and rowing [33].

While these methods can be applied to virtually any sport, others used sensor placement or data processing algorithms more specific to a given assessment context. For example, Gurchiek, McGinnis [30] successfully used an adaptive filter iteratively correcting accelerometer-based velocity estimates to ensure a proper fit to a theoretical model of speed modulations during maximal effort sprint. Regarding the sensor placement, an IMU on the foot allowed for the use of the zero-velocity update method to correct the drift during each stance phase, when the foot velocity was expected to be null [34]. During wheelchair sports, a common sensor placement to measure the wheelchair velocity is on the wheel. In these studies, instead of integrating accelerometer data, the angular velocity was extracted directly from the IMU gyroscope (after sensor calibration) and was multiplied by the wheel’s radius to estimate the forward velocity [21,22,23,24,25]. While this method has been proven useful in wheelchair sports, we chose to assess wheelchair speed with an IMU placed on the frame for three main reasons. First, the wheelchair frame offers a large and stable base for the hardware placement. This facilitates the implementation of the algorithms on various wheelchair models and measurement hardware (e.g., an embedded smartphone IMU). Second, frame-based IMUs were proven superior to wheel-based IMUs to detect the hand strike during WR [16]. Last, frame-based IMUs have the potential to provide information about changes in the wheelchair heading direction (the rotation around the vertical axis), providing the opportunity to assess the behavior during curves and overtaking. While these points are beyond the scope of this study, the validation of a frame-based accelerometer to assess the wheelchair speed is a first step to the development of a WR feedback system.

## 3. Materials and Methods

### 3.1. Participants

Thirteen able-bodied participants without experience in WR (twelve males and one female; age = 23 ± 1; body mass index [BMI] = 24 ± 3; referred to as novice participants [NPs] for the rest of the manuscript) and three wheelchair racing athletes (WRAs) (competing in sport classes T51, T52, and T54; three males; age = 26 ± 6) were recruited with informed consent among Université Laval students and wheelchair racing athletes competing in the province of Quebec. NPs were recruited through the Université Laval mailing list and on-campus publicity. WRAs were recruited with the help of a provincial parasports federation (i.e., Parasports Québec) and of N.S., who is the parasport coach within the university athletic club. Among the WRAs, two (WRA1 and WRA3) were competing at international events at the time of data collection, and one (WRA2) was competing at national events. The sample size was selected according to similar studies [26,27]. Based on previous studies [25], ICC values of around 0.85 were expected during the reliability analyses. It was estimated that 11 participants would be required to ensure that the 95% confidence interval of the ICC had a width ≤ 0.35, thereby conservatively maintaining the confidence interval within the moderate reliability category (ICC ≥ 0.5) [35]. Such a sample size would also ensure that the strong correlations (r ≥ 0.7) obtained during the convergent validity analyses would be statistically significant (GPower, α = 0.95 and β = 0.8). NPs were included if they were nonsmokers, aged between 18 and 30, had a body mass index (BMI) < 30, and were free from a disorder or pain that could affect the task. All of the subjects gave their informed consent for inclusion before they participated in the study. The study was conducted in accordance with the Declaration of Helsinki, and the study protocol was approved by Université Laval’s ethics committee (#2022-077).

### 3.2. Instrumentation

The NPs used the same standardized wheelchairs, with the tires inflated at 120 psi, and soft gloves (Figure 1). The WRAs used their own competition wheelchair with their preferred gloves and tire inflation settings. NPs performed on the indoor track of Université Laval. As the WRAs live all over the province of Quebec, they were evaluated on the outside track of their choice. One Lidar system (SpeedTracker, SciencePerfo, Québec, QC, Canada; 50 Hz) was positioned on an adjustable tripod 4 m behind the starting line. The height of the system was aligned to detect a reflective plate fixed on the wheelchair rear bars of the frame. Considering the ecological aspect of the data collection on an athletic track, a 3D camera system was not used as a gold standard for time and space constraints. The Lidar system, despite not being a gold standard, was chosen for its ease of use and for the high validity and reliability of similar radar and laser devices [36]. One IMU (Physilog, Mindmaze, Lausanne, Switzerland; 256 Hz, 16 g), running continuously, was positioned on the rear horizontal bar of the wheelchair frame using a Velcro elastic band. All of the data collections were carried out by the same author (R.O.).

### 3.3. Acquisition Protocol

The NPs performed one familiarization (T0) and two data collection sessions (T1 and T2), each separated by one week at the same time of the day. The collection sessions were all set in the evening with a reserved corridor to avoid the busy hours of the indoor track during summer. The WRAs only performed one data collection session (T1). These data collection sessions were performed on a reserved corridor during a calm, windless day, after a rest day. The participants were asked to avoid any unusual exercise 24 h before the testing sessions. At T0, the participants were introduced to the propulsion technique before practicing linear accelerations on the track until verbal notice of familiarity. At T1 and T2, following the placement of the IMU, both a static pose (static calibration along the vertical axis) and tilt motions around the wheel axis (dynamic calibration along the mediolateral axis) were recorded. This was performed to mathematically realign the IMU with the wheelchair reference frame. For the WRAs, the IMU was reoriented using the static pose only as they remained seating in their wheelchair. The mediolateral axis was estimated with the horizontal bar of the frame, where the IMU was positioned. The NPs then performed a 15-min progressive warm-up of linear accelerations. The warm-up of the WRAs was the same as the one they usually performed before a maximal effort training workout. Afterward, they performed static start maximal effort accelerations for a 36 m distance on an auditory signal. The trials were carried out in succession with complete rest (3-min minimum) until the completion of 3 valid trials per session. A trial was considered invalid if the participant crossed the corridor before reaching the distance and if the user was destabilized by a wheelie (the front wheel of the wheelchair no longer touching the ground) during the trial.

### 3.4. Data Processing

All of the data analyses were completed with a custom-built Matlab (version R2021b, Mathworks, Natick, MA, USA) program. The IMU acceleration and angular velocity data, and the Lidar position data, were low-pass filtered with a cut-off frequency of 4 Hz. The IMU accelerometer data were reoriented with the wheelchair frame of reference based on calibration trials. The unit vector pointing in the anteroposterior direction (x_IMUap_, y_IMUap_, and z_IMUap_) was identified as the cross-product between the mean acceleration unit vector (x_IMUv_, y_IMUv_, and z_IMUv_) and the range of angular velocity unit vector (x_IMUml_, y_IMUml_, and z_IMUml_) collected during the static and tilt motion calibration recordings, respectively. A rotation matrix (^wheelchair^R_IMU_) was then computed and used to reorient the accelerometer data on the anteroposterior (x_wheelchair_), vertical (y_wheelchair_), and mediolateral (z_wheelchair_) axes of the wheelchair using Equations (1) and (2), as follows:(1)xwheelchairywheelchairzwheelchair=RIMUwheelchairxIMUyIMUzIMU(2)RIMUwheelchair=100010001xIMUapxIMUvxIMUmlyIMUapyIMUvyIMUmlzIMUapzIMUvzIMUml′=xIMUapxIMUvxIMUmlyIMUapyIMUvyIMUmlzIMUapzIMUvzIMUml′

The wheelchair speed was calculated as the differentiation of the Lidar system position. As for the IMU system, a preliminary (drifting) speed estimation was computed using the integration of the IMU anteroposterior acceleration (x_wheelchair_). Both systems were then synchronized using a cross-correlation function applied to the product of the acceleration (i.e., the double-differentiation of the position signal for the Lidar system) and computed speed. An incrementation strategy, similar to Clement et al. (2021) [32], was then used to correct for the drift introduced by the accelerometer integration. Specifically, the IMU acceleration signal was incrementally adjusted by 0.001 m/s^2^ until the 30-m time estimated from its double integration was within the arbitrary window of 0.01 s of agreement with the Lidar system. Final IMU-based velocity estimates were computed through the integration of this adjusted acceleration signal. Then, for each trial, data were segmented into individual propulsion cycles (PCs) using the local minima of the IMU forward acceleration. Data segmentation was visually inspected and corrected if needed. The outlying PCs (e.g., because of a slip) were rejected and not taken into consideration for the average.

### 3.5. Metric Extraction

The root mean square difference between the Lidar and IMU speed curves (RMSe) was computed for each PC. The normalized RMSe was expressed as a percentage of maximal speed reached after 10 PCs. Multiple wheelchair kinematic and spatiotemporal metrics were simultaneously extracted from the IMU and Lidar system data for each PC, as shown on Figure 2 and Table 1.

### 3.6. Statistical Analysis

For the NPs, statistical analyses were carried out on the average of the 3 trials for the first push (PC1) and the mean of PC3 to 10 (PCstable) using Jamovi software v 2.3.18 [37]. T1 and T2 were compared for both systems using a two-way mixed, absolute agreement, intraclass correlation coefficient model for the average measurement (ICC 3,1). The results were interpreted as suggested by Koo and Li [38], as follows: ICC < 0.5 as poor, 0.5 ≤ ICC < 0.75 as moderate, 0.75 ≤ ICC < 0.90 as good, and ICC ≥ 0.90 as excellent. The standard error of measurement (SEM) and minimal detectable change (MDC) were computed using Equations (3) and (4), respectively, as follows:(3)$$SEM=SD\times \sqrt{1-ICC}$$(4)$$MDC=SEM\times 1.96\times \sqrt{2}$$
where SD is the standard deviation of the observed scores [39]. The convergent validity between the two systems at T1 was analyzed using the Pearson correlation coefficient (r) and was interpreted as suggested by Mukaka [40], as follows: low (r ≤ 0.49), moderate (0.50 ≤ r ≤ 0.69), high (0.70 ≤ r ≤ 0.89), and very high (0.90 ≤ r). The root mean square error (RMSe) was also computed for every propulsion cycle at T1 for the NPs. The t-tests and Cohen’s D (mean difference/pooled SD) effect size statistics were conducted between T1 and T2, and between both systems at T1, to detect significant systematic biases between sessions or systems. An alpha (α) of 0.05 was accepted for significance. For the secondary objective, descriptive statistics for the IMU-based metrics were visually compared between the WRAs and NPs at T1. Furthermore, the rank of the WRAs among the whole sample was computed for each metric. Figures were produced with Matlab (version R2021b, Mathworks, Natick, MA, USA) and Graphpad Prism Pro 10 (Graphpad software LLC, Boston, MA, USA) software.

## 4. Results

### 4.1. Convergent Validity

The RMSe between the IMU-based and Lidar-based velocity curves was 0.16 ± 0.03 m/s (4.62 ± 0.69%) and 0.30 ± 0.09 m/s (9.25 ± 2.91%) for PC1 and PCstable, respectively (Figure 3).

At PC1, high positive correlations for AccMax and AvSpeed/AvSpeedGain, and a very high positive correlation between systems for AccGain, were obtained (see Table 2 for descriptive statistics; see Table 3 for inferential statistics). For PCstable, high positive correlations for AvSpeed and AvSpeedGain, and very high positive correlations for AccMax, AccMin, AccGain, and DecLoss, were obtained. No significant correlations were found for AccMin at PC1. At PC1, the Lidar system values were significantly higher than the IMU for AccMax, but significantly lower for AvSpeed and AvSpeedGain. For PCstable, the Lidar system values were significantly higher than the IMU for AccMax.

### 4.2. Test–Retest Reliability

The IMUs showed moderate-to-good reliability for the PC duration and frequency at PC1 and PCstable (see Table 1 for descriptive statistics; see Table 2 for inferential statistics). There were no significant differences between the two sessions for the spatiotemporal metrics.

As for the wheelchair kinematics, the IMUs showed moderate-to-good reliability for AccMax and AvSpeed/AvSpeedGain at PC1. Poor reliability was found for AccGain and AccMin at PC1. The IMUs showed significant moderate-to-good reliability for the AccMax, AccMin, AccGain, and DecLoss metrics at PCstable. No significant ICC was found in AvSpeed and AvSpeedGain at PCstable. There were no significant differences between T1 and T2 for all of the studied metrics.

The Lidar system showed moderate-to-good reliability for the AccMax and AvSpeed/AvSpeedGain metrics at PC1. No significant correlation was found in the other metrics at PC1. The Lidar system showed good reliability for all of the studied metrics at PCstable. There were no significant differences between T1 and T2 for all of the studied metrics.

### 4.3. Comparison of NPs and WRAs

The three WRAs reached a greater speed than all of the NPs during both PC1 and PCstable (Figure 4A,B). The WRAs were among the best-ranked participants for the speed generation (e.g., AccGain and AccMax; WRA2-3) or deceleration (DecLoss and AccMin; WRA1-3) metrics (Figure 3B). Regarding the spatiotemporal parameters, WRA1 and WRA2 used a relatively long propulsion cycle duration (i.e., low frequency), while WRA3 used a short propulsion cycle duration (i.e., high frequency).

## 5. Discussion

The objective of this study was to evaluate the test–retest reliability and concurrent validity of wheelchair racing performance metrics on abled-bodied individuals. For our first objective, we expected good-to-excellent reliability (ICC > 0.80) and high correlations (r ≥ 0.70) between the IMU and Lidar systems. For our second objective, we expected the WRAs to perform better than the NPs according to most of the extracted metrics. Our results show moderate-to-good reliability for the IMU and Lidar systems, as well as high correlations between the systems for almost all of the metrics. Furthermore, the comparison between the WRAs and NPs also highlighted the sensitivity of the measures.

### 5.1. Test–Retest Reliability

When looking at the test–retest reliability, our results show a good-to-moderate ICC for most (22/28) of the analyzed metrics, which is slightly less than expected. The ICC values found for most of the metrics are similar to the test–retest reliability of the 12-m sprint time in elite wheelchair basketball (ICC = 0.62), attesting that sprinting from a still position is a hard task to replicate even for experienced wheelchair users [41]. The developed metrics quantified parts of a PC, not only the overall performance over a task. This precision may be more sensitive to changes in the position or technique between sessions. Moreover, the described ICC values could be affected by the characteristics of the systems, but also by the trial-to-trial consistency and the between-subject variability [39]. For example, the metrics based on the acceleration signal have a lower reliability for the Lidar system than for the IMU. This could be explained by an error due to the double differentiation of the signal, which is known to greatly amplify the measurement noise. On the other hand, the IMU has a lower reliability for average speed metrics. This may indicate that the implemented signal processing strategy did not completely remove the accumulation of errors (i.e., drift) inherent to the integration of the accelerometer signal over multiple PCs [42,43,44]. Other metrics at PC1, such as AccGain, could be affected by the inertia of the wheelchair–user system to overcome, which creates higher technical variability, especially for novice participants [3,7]. Nonetheless, our results suggest that the IMU and Lidar systems would be sufficiently reliable for detecting the differences in performance over the 3-week training plan of untrained participants for the cycle duration (MDC = 0.08) and AvSpeed (MDC = 0.47) of the participants in the de Klerk et al. (2022) [45] study. In the end, the metrics are reliable enough to support their use for assessing the WR performance in novice participants over two different periods. Feedback based on such metrics can surely support the initial exploration of propulsion techniques and equipment adjustments, and contribute to the motivation of individuals newly involved in the sport, by showing their progression in the different parts of the race.

### 5.2. Convergent Validity

A comparison of the velocity signal derived from the IMU and Lidar systems revealed a convergent validity comparable to previous studies conducted in other sport contexts. Specifically, the RMSe values obtained in the current study were comparable to those obtained when using velocity computation algorithms based on accelerometer integration during paraswimming, sprint running, or rowing (Table 4). The developed speed metrics had high to very high correlations with its comparative, the Lidar system. Moreover, at PCstable, there was no significant difference between both systems for all speed metrics. The Lidar system, despite not being a gold standard, is a system known for its high validity and reliability [36]. However, for PC1, AvSpeed/AvSpeedGain were significantly greater in favor of the Lidar system. As for the acceleration metrics, AccMax had high (PC1) to very high (PCstable) correlations between systems, but was significantly greater when measured with the IMU. AccMin had no significant correlations between the systems for all of the analyzed PCs, and were statistically different between both systems. These differences could be explained by the strengths and weaknesses for each system, as discussed for the reliability analysis. Each system has differences in their data processing, which can result in an overestimation for the speed or acceleration. Overall, the results for PCstable and the high correlations confirm our first hypothesis for the convergent validity. This convergence between systems supports the use of the various speed metrics developed either with an IMU or a Lidar system.

### 5.3. Comparison with WRAs

The NPs included in our study reached an average speed (~2.4 m/s) similar to the NP participants of de Klerk et al. (2022) (~2.34 m/s) [45]. On the other hand, they had a much lower average speed and average speed gain than the WRAs included in our study and in previous studies (≥5 m/s), as expected [4]. These results highlight the sensitivity of the metrics to differentiate participants with different performance classes. For the other metrics, they are greatly affected by the propulsion technique. Effectively, the use of the trunk, the patterns of the upper body during the recovery phase, and the emphasis on either having a lower push, but faster recovery, while keeping a more aerodynamic position are many of the variables which may influence the other developed metrics. This is shown by the mixed ranks amongst all of the participants and the high standard deviation. Likewise, the performance of the WRAs in our study was heterogeneous due to their different classifications, and, therefore, different capacities. For instance, T51 athletes do not have strength in trunk muscles, and tend to have difficulties with the elbow extension, resulting in a propulsion generated by elbow flexion [46]. This propulsion strategy used by WRA3 was visually different from the other WRAs and NPs during data collection. Given this large heterogeneity, using within-subject designs to analyze variations in individual performance may be a suitable option to understand the factors influencing speed generation and speed loss in future studies. Such a strategy could also contribute to the development of individualized training programs.

### 5.4. Limitations

A few limitations must be taken into consideration. First, the sample size used for the statistical analysis is small and mostly consists of participants who have different physical capabilities and experience from wheelchair athletes, which can influence the reliability [47,48]. Since the reliability is sensible to the population, the results must be interpreted with caution and cannot be used for experienced users. Furthermore, the use of a larger sample size would improve the precision of the validity and reliability statistics (smaller confidence intervals). Second, the analyzed PCs cover a distance which is smaller than the 100-m race, the shortest event in WR. Using a more representative effort could maximize the usability of the developed metrics for guiding the training of WRAs. Future studies could perform sensitivity analyses to better understand the impact of the race distance on the stability of the IMU-based measures. Such a study could also assess the impact of alternative, more usable, methods to correct for the IMU drift based on the time required to reach any known distance (e.g., timing chips or a manual chronometer). Third, the Lidar system is not a gold standard for speed analysis, but the use of similar technology is encouraged in that regard [36]. Finally, most of the NPs and all of the WRAs were men. Despite these limitations, this study gives a first idea of the robustness of the metrics.

## 6. Conclusions

This study supports the use of single-frame-based IMUs and Lidar systems for assessing WR performance, particularly in novice participants. The comparison between the WRAs and NPs highlighted the sensitivity of the metrics to different performance levels and techniques. Despite some limitations, such as the small sample size and variability in the participant capabilities, the results provide a robust foundation for future research. Further studies on male and female para-athletes, and over longer distances, are needed to maximize the benefits of the system.

## Figures and Tables

**Figure 1 sensors-25-01680-f001:**
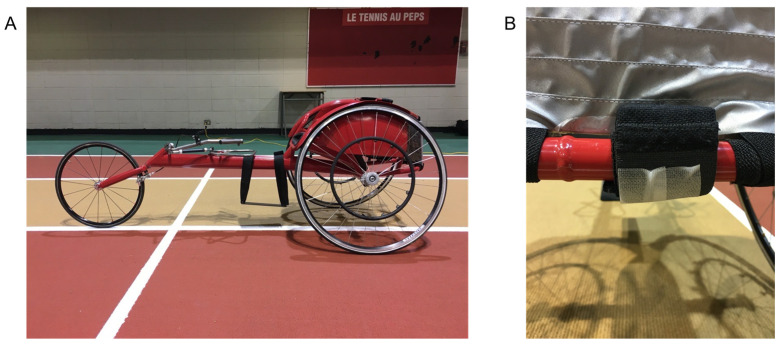
(**A**) Racing wheelchair used for the NPs. (**B**) Inertial measurement unit and the reflective plate positioned on the back of the frame.

**Figure 2 sensors-25-01680-f002:**
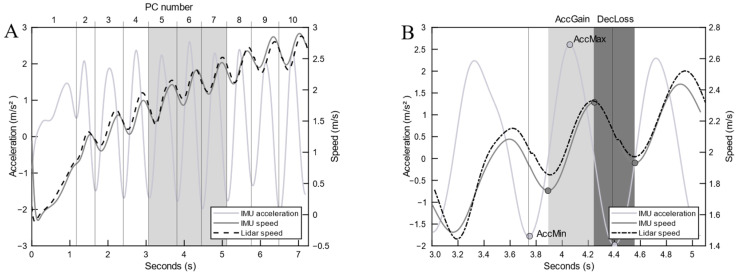
Cycle separation (**A**) and metric extraction (**B**). Panel A: Gray area highlights the propulsion cycles 5 to 7, which are illustrated in panel B. Panel B: Light-gray area highlights the acceleration phase (AccGain); dark-gray area highlights the deceleration phase (DecLoss). AccMin: minimal acceleration; AccMax: maximal acceleration; AccGain: speed gains during the acceleration phase; DecLoss: speed loss during deceleration.

**Figure 3 sensors-25-01680-f003:**
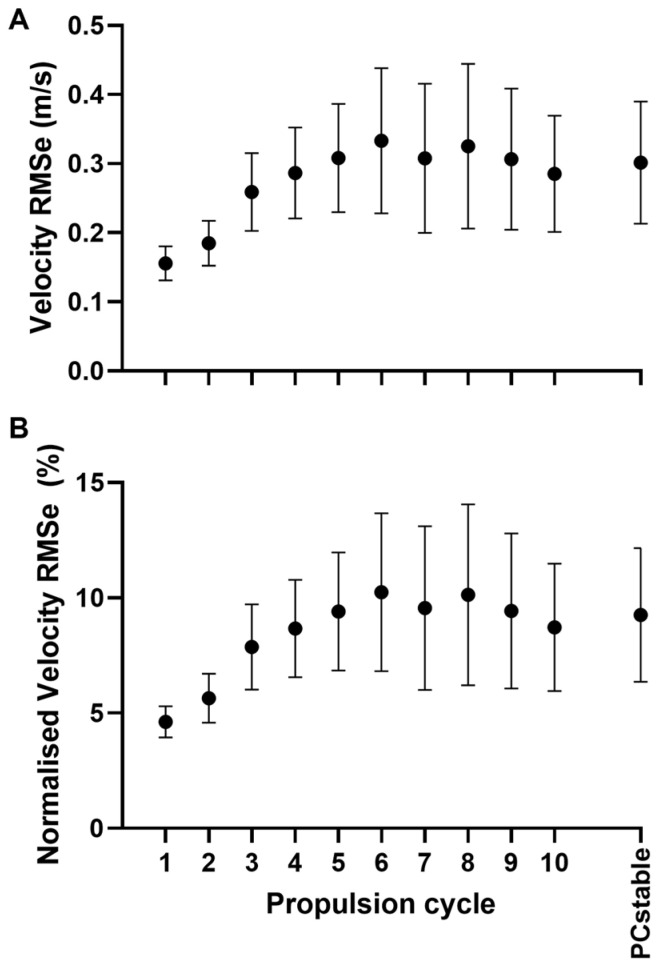
(**A**) Absolute velocity RMS error (RMSe); (**B**) velocity RMS error normalized to the maximal velocity over the ten first propulsion cycles. PCstable: propulsion cycles 3 to 10.

**Figure 4 sensors-25-01680-f004:**
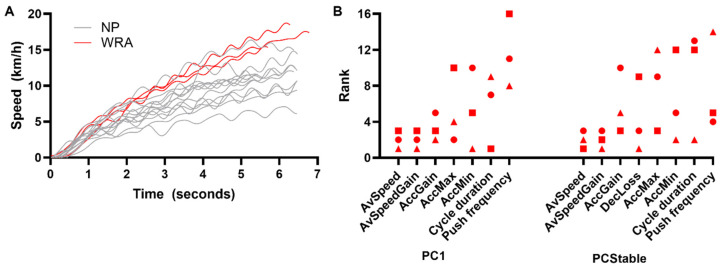
(**A**): Wheelchair velocity curves of all NPs (gray) and WRAs (red). (**B**): Ranks of the WRAs within the entire group (for a total of 16 participants: 3 WRAs + 13 NPs). 
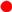
 WRA1, 
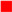
 WRA2, and 
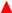
 WRA3. AccMin: minimal acceleration; AccMax: maximal acceleration; AvSpeed: average speed; AvSpeedGain: average speed gain between two cycles; AccGain: speed gains during the acceleration phase; DecLoss: speed loss during deceleration; PC1: first propulsion cycle; PCstable: propulsion cycles 3 to 10.

**Table 1 sensors-25-01680-t001:** Spatiotemporal and wheelchair kinematic metrics extracted for each PC.

Metric	Definition	Unit
Cycle duration	Time difference between the beginning and end of a PC	s
Cycle frequency	Reciprocal of the cycle duration	Stroke/s
Maximal acceleration (AccMax)	Maximal value of the anteroposterior acceleration within a PC	m/s^2^
Minimal acceleration (AccMin)	Minimal value of the anteroposterior acceleration within a PC	m/s^2^
Average speed (AvSpeed)	Average anteroposterior velocity during a PC	m/s
Average speed variation (AvSpeedGain)	Changes in the AvSpeed in the current PC relative to the preceding PC *	m/s
Velocity gain during the acceleration phase of the PC (AccGain)	Difference between the maximal velocity reached during a PC and the velocity at the beginning of the PC	m/s
Velocity loss during the deceleration phase of the PC (DecLoss)	Difference between the velocity at the end of the PC and the maximal velocity reached during a PC **	m/s

* By definition, AvSpeedGain = AvSpeed for PC1. ** DecLoss was not analyzed for PC1.

**Table 2 sensors-25-01680-t002:** Descriptive statistics for the IMU and Lidar systems at T1 and T2.

			T1 IMU	T2 IMU	T1 Lidar	T2 Lidar
		Units	Mean ± SD	Mean ± SD	Mean ± SD	Mean ± SD
PC1	Cycle duration	s	1.01 ± 0.25	1.03 ± 0.23	-	-
	Push frequency	Hz	1.07 ± 0.29	1.04 ± 0.31	-	-
	AccMax	m/s^2^	1.75 ± 0.33	1.70 ± 0.44	2.09 ± 0.41	2.15 ± 0.41
	AccMin	m/s^2^	−0.34 ± 0.26	−0.27 ± 0.33	−0.44 ± 0.66	−0.32 ± 0.33
	AvSpeed	m/s	0.42 ± 0.1	0.44 ± 0.13	0.36 ± 0.12	0.39 ± 0.14
	AvSpeedGain	m/s	0.42 ± 0.1	0.44 ± 0.13	0.36 ± 0.12	0.39 ± 0.14
	AccGain	m/s	1.06 ± 0.26	1.05 ± 0.30	1.05 ± 0.30	1.09 ± 0.31
PCstable	Cycle duration	s	0.57 ± 0.08	0.53 ± 0.08	-	-
	Push frequency	Hz	1.81 ± 0.28	1.94 ± 0.30	-	-
	AccMax	m/s^2^	1.69 ± 0.60	1.70 ± 0.56	1.76 ± 0.55	1.84 ± 0.51
	AccMin	m/s^2^	−1.04 ± 0.59	−0.99 ± 0.53	−0.99 ± 0.59	−0.98 ± 0.47
	AvSpeed	m/s	2.40 ± 0.36	2.40 ± 0.35	2.36 ± 0.33	2.42 ± 0.33
	AvSpeedGain	m/s	0.25 ± 0.05	0.25 ± 0.05	0.24 ± 0.03	0.25 ± 0.04
	AccGain	m/s	0.37 ± 0.12	0.35 ± 0.08	0.35 ± 0.11	0.34 ± 0.09
	DecLoss	m/s	−0.15 ± 0.11	−0.12 ± 0.07	−0.13 ± 0.11	−0.11 ± 0.07

AccMin: minimal acceleration; AccMax: maximal acceleration; AvSpeed: average speed; AvSpeedGain: average speed gain between two cycles; AccGain: speed gains during the acceleration phase; DecLoss: speed loss during deceleration; PC1: first propulsion cycle; PCstable: propulsion cycles 3 to 10.

**Table 3 sensors-25-01680-t003:** Results of the statistical analyses for the convergent validity and reliability.

			Reliability Analyses (T1 vs. T2)					Validity Analyses (Lidar vs. IMU; T1)	
			ICC	ICC CI	SEM	MDC	t-Test *p*-Value; Cohen’s D	t-Test *p*-Value; Cohen’s D	Correlation: r; *p*-Value
PC1	AccMax	IMU	0.86	0.56–0.95	0.13	0.27	0.53; −0.13	0.001; 0.92	0.78; 0.002
		Lidar	0.68	0.08–0.90	0.23	0.45	0.76; 0.15		
	AccMin	IMU	[0.47]	[5.38]–0.58	0.35	0.97	0.61; 0.24	0.55; −0.22	0.40; 0.17
		Lidar	0.57	[0.43]–0.87	0.34	0.94	0.47; 0.24		
	AvSpeed	IMU	0.67	[0.08]–0.9	0.07	0.13	0.49; 0.17	0.01; −0.55	0.86; <0.001
		Lidar	0.80	0.36–0.94	0.1	0.19	0.9; 0.23		
	AvSpeedGain	IMU	0.67	[0.08]–0.9	0.07	0.13	0.49; 0.17	0.01; −0.55	0.86; <0.001
		Lidar	0.80	0.36–0.94	0.1	0.19	0.9; 0.23		
	AccGain	IMU	0.40	[1.21]–0.823	0.21	0.42	0.93; −0.04	0.42; −0.04	0.98; <0.001
		Lidar	0.20	[1.88]–0.75	0.28	0.56	0.94; 0.13		
	Cycle duration	IMU	0.67	[0.14]–0.901	0.13	0.27	0.75; 0.08		
	Push frequency	IMU	0.72	0.03–0.91	0.16	0.31	0.75; −0.1		
PCstable	AccMax	IMU	0.81	0.36–0.94	0.25	0.49	0.99; 0.02	0.047; 0.12	0.99; <0.001
		Lidar	0.77	0.28–0.93	0.25	0.49	0.45; 0.53		
	AccMin	IMU	0.86	0.53–0.96	0.21	0.57	0.68; 0.09	0.30; −0.08	0.97 <0.001
		Lidar	0.87	0.56–0.96	0.19	0.52	0.93; 0.02		
	AvSpeed	IMU	0.52	[0.71]–0.86	0.24	0.47	0.98; 0.00	0.5; −0.12	0.83; <0.001
		Lidar	0.78	0.32–0.93	0.15	0.3	0.53; 0.33		
	AvSpeedGain	IMU	0.49	[0.87]–0.85	0.03	0.06	0.92; 0.00	0.64; −0.25	0.79; 0.001
		Lidar	0.77	0.32–0.93	0.02	0.04	0.12; −0.04		
	AccGain	IMU	0.77	0.26–0.93	0.05	0.1	0.36; −0.2	0.09; −0.17	0.95; 0.001
		Lidar	0.78	0.29–0.93	0.05	0.09	0.9; 0.10		
	DecLoss	IMU	0.77	0.28–0.93	0.04	0.09	0.28; 0.33	0.1; 0.18	0.96; < 0.001
		Lidar	0.78	0.33–0.93	0.04	0.08	0.51; 0.09		
	Cycle duration	IMU	0.74	0.17–0.92	0.04	0.08	0.08; −0.50		
	Push frequency	IMU	0.81	0.35–0.94	0.13	0.25	0.06; 0.44		

AccMin: minimal acceleration; AccMax: maximal acceleration; AvSpeed: average speed; AvSpeedGain: average speed gain between two cycles; AccGain: speed gains during the acceleration phase; CI: confidence interval; DecLoss: speed loss during deceleration; PC1: first propulsion cycle; PCstable: propulsion cycles 3 to 10.

**Table 4 sensors-25-01680-t004:** Comparison of the convergent validity obtained in the current study with previously published research.

Study	Sport	Algorithm	Reference System	RMSe Stable (m/s; % Velocity)
Current	Wheelchair racing	Integrated acceleration with the drift corrected with time at 30 m	Lidar system (Speedtracker, SciencePerfo)	0.30; 9.25%
[32]	Paraswimming	Integrated acceleration with the drift corrected every 50 m	Cable linear transducer (1080 Sprint, 1080 Motion)	Freestyle: 0.14; 12.6% Butterfly: 0.36; 31.3% Breaststroke: 0.39; 46.4% Backstroke: 0.16; 17.0%
[33]	Rowing	Complementary (CF) or Kalman (KF) filters to combine integrated acceleration with smartphone GPS	Differential GPS (model unknown)	CF: Elite: 0.33; 7.8% Club-level: 0.31; 7.9% KF Elite: 0.27; 6.4% Club-level: 0.32; 8.2%
[31]	Sprint running	Kalman filter combining integrated acceleration with GPS, considering a sprint velocity theoretical model	Radar system (ATS PRO II, Stalker Sport)	60 m: ≈ 0.65 m/s; 6.5%

## Data Availability

The original contributions presented in this study are included in the article; further inquiries can be directed to the corresponding author.

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
