# Peer review of "Validity and Reliability of Inertial Motion Unit-Based Performance Metrics During Wheelchair Racing Propulsion"

_sensors, 2025, doi:10.3390/s25061680_

Round 1

Reviewer 1 Report

Comments and Suggestions for Authors

GENERAL COMMENTS

The aim of this paper was to evaluate the concurrent validity of the Inertial Measurement Unit (IMU) compared to the Lidar system for assessing wheelchair racing performance metrics, as well as to test the test-retest reliability of these measures. While this article addresses an intriguing and relevant topic, several issues must be resolved before it can be considered for publication.

SPECIFIC COMMENTS

ABSTRACT

  • The term "IMU" should be clearly defined within the abstract for the benefit of readers who may not be familiar with it. Additionally, I suggest adding a line discussing the implications of the findings to provide a more comprehensive overview of the study's significance.

INTRODUCTION

  • The introduction requires major revision and clarification. Currently, the aim of the study is not clearly articulated. It is essential to explain why comparing the IMU to the Lidar system, as well as assessing the test-retest reliability of wheelchair racing performance metrics, is important. What significance does this study hold for the field?
  • Research manuscripts should answer specific research questions; however, this manuscript lacks clarity in that regard. The authors should restructure the introduction to guide readers through the context and rationale for their research. By providing a clear background on the relevance of this study, the authors can lead to a well-defined research question. Unfortunately, the introduction is neither concise nor sufficiently informative about the importance of the study.

METHODS

  • The methods section requires some revisions:
    1. Please clarify the rationale for including 12 male participants and only 1 female participant. Is there a specific reason for having only one female participant? Consider whether this approach serves the study's goals and whether it may be appropriate to evaluate a more balanced sample.
    2. Please clarify what "NP" stands for, as this is not defined within the text.
    3. Include a sample size calculation to provide justification for the chosen sample size. The current sample size raises concerns about the statistical power of the study.
    4. The procedure for recruiting participants should be included to enhance transparency. How were participants selected, and what criteria were used?

DISCUSSION

  • In the discussion section, the authors need to expand on their findings and the implications of these results. A deeper exploration of how these findings contribute to the field of wheelchair racing performance would strengthen this section.
  • Additionally, the authors discuss several topics unrelated to the study's results, which can detract from the core findings. While it is important to reference other studies, many details provided may not be necessary for the discussion.
  • The limitations of the study are not adequately addressed. Please revise this section to thoughtfully consider and discuss the limitations, providing a more comprehensive perspective on how they might impact the findings.

Thank you for considering these revisions. I look forward to seeing the improvements made to this manuscript.

Comments on the Quality of English Language

OK.

Reviewer 2 Report

Comments and Suggestions for Authors

The purpose of this study was to assess the concurrent validity (IMU vs Lidar system) and test-retest reliability of wheelchair racing performance indicators. Persons without disabilities and wheelchair racing experience were assessed twice for the maximum short distance effort in the racing wheelchair. Three wheelchair athletes were also evaluated to compare their performance with novices. Using simultaneous estimation of wheelchair kinematics, the IMU and LIDA propulsion cycle (PC) duration, acceleration, average velocity, speed gain at acceleration and speed loss at deceleration were estimated and compared the difference between the two systems in the first PC and stable PC. Of the indicators analyzed, most showed test-retest reliability of moderate (0.50 ICC <0.75) to good (0.75 ICC <0.90), while five of the indicators had poor reliability (ICC <0.50). Both systems achieved high to very high correlations between 9 of the 11 indicators (0.78-0.99), indicating that they are accurate and supporting their use over short field distances. Wheelchair athletes perform better than novices. The findings of this paper suggest that integrated accelerometer data can be used to assess the velocity properties of the wheelchair over short distances of known transit time.

We affirm the contribution of this paper, and at the same time, there are some problems and deficiencies in the paper also need to be pointed out to improve.

1.The explanation of the proposed method is not detailed enough, we should focus on the interpretation of the method proposed in the paper, and describe as much detail as possible, so as to help readers to understand the method of the paper.

Please refer the recent papers: Chenggang Yan, Biao Gong, Yuxuan Wei, Yue Gao, “Deep Multi-View Enhancement Hashing for Image Retrieval”, IEEE Transactions on Pattern Analysis and Machine Intelligence, 2020.

2.The structure of the paper is lacking, for example, the contribution made to the paper should be listed in the text, so that readers can better grasp the core contribution of the paper, and in addition, the focus of the paper should be explained.

3.The relevant ablation experiments in the article were not done enough, for the research depth of the proposed algorithm is lacking, should be for the various parts of the proposed algorithm ablation experiment has determined what part for the results of what contribution, it will make the paper algorithm more convincing, more profound for the understanding of the algorithm.

4.The article typesetting is not beautiful enough, leaving more white space.

5.There are too few formulas in the paper, so some more formulas are needed to describe the calculation principles.

Comments on the Quality of English Language

none

Reviewer 3 Report

Comments and Suggestions for Authors

About the work:

The paper presents a not very clear approach regarding the use of inertial sensors and a lidar system. There is a lack of organization in the state-of-the-art approach (a clear section on this is suggested). The working methodology is not very clear, and the entire procedure needs to be better described. There is a lack of theoretical description. The results are not clear or organized. Finally, there is no clear conclusion section (which is combined with the discussion of the results).

More specific details follow:

1) The acronyms should all be presented beforehand, even in the abstract

See in line 10 : (IMU vs Lidar system) --> (inertial vs Lidar systems)

In line 18: ICC ?

In line 68: WRA (first time in line 68, but it was defined only in line 74)

In line 73: BMI

== > Perhaps you could insert an “Acronym, abbreviation sumary” before the References section

ICC does not appear in the entire article:

==> “ICC is an intraclass correlation coefficient to perform test/retest reliability, in another words, ICC is a reliability index that reflects both degree of correlation and agreement between measurements.”

Round 2

Reviewer 3 Report

Comments and Suggestions for Authors

Re-review of “sensors-3241315”, entitled: “Validity and Reliability of Inertial Motion Units-Based Performance Metrics During Wheelchair Racing Propulsion”

There were 22 comments. I was surprised to see that, in this return, part of the content appeared for the authors (only item 1) and another part (from item 2 to 22) exclusively for the editors... My argument was complete and justified the suggested rejection, not just on the basis of the 4 points answered by the reviewers, but on the basis of a whole set of arguments. Please, editors, check what I am saying so that there is no problem of interpretation or discomfort with the authors.

Anyway, 

I'll continue with the original comments below (From 1 to 22) and after them, I'll give a brief assessment of the changes made so far.

=========================

REVIEW 1

=========================

The work needs to be better organized and the topic better approached. A review of the state of the art will also be important. In terms of sensors, there is little description and a lack of technical approach.

1) The acronyms should all be presented beforehand, even in the abstract

See in line 10 : (IMU vs Lidar system) --> (inertial vs Lidar systems)

In line 18: ICC ?

In line 68: WRA (first time in line 68, but it was defined only in line 74)

In line 73: BMI

== > Perhaps you could insert an “Acronym, abbreviation sumary” before the References section

ICC does not appear in the entire article:

==> “ICC is an intraclass correlation coefficient to perform test/retest reliability, in another words, ICC is a reliability index that reflects both degree of correlation and agreement between measurements.”

I think there is a need to revise the references. Many are only cited, without being addressed. And others are missing, as I described in my review report.

The work needs to be better organized and the topic better approached. A review of the state of the art will also be important. In terms of sensors, there is little description and a lack of technical approach.

I don't think it meets the criteria of Sensors readers, as it is technically weak and often confusing.

2) It is necessary to include many references in the line33 statement.

3) I suggest that the authors address the state of the art more clearly, and perhaps separately. I suggest inserting a table of related work at the end of this section, with comparative characteristics, showing the relevant results and limitations of the work, so that the contribution of this work is clearly contextualized.

4) It is important that the authors deal technically with all the references used. For example: 

a) “...IMU-based kinematics metrics showed good validity [11-15]”

b) “...and intra-test reliability [11, 16, 17]”.

c) “In addition, the majority of studies relating to WCS and WR use a wheel-based configuration of 3 IMUs, ... [14, 18-22]”

==> some of the references in these excerpts (such as 12, 13 and 20) were not even addressed, but only appeared as “related to the previous statement”.

5) The text is very confusing.

e.g. (WRA) (T51, T52, T54; 3 men; age = 26 ± 6)

Within the WRA, two (WRA1, WRA3)... and one (WRA2)

==> TRANSLATION: PT-SUBS TEAM SYNCHRONIZATION: PT-SUBS TEAM

==> What is the T** nomenclature? It seems to refer to WR*... but why two names?

==> Translation: PT-Subs Team Synchronization: PT-Subs Team

6) “The sample size was selected according to similar studies” --> ok, but it's very, very important to describe the “experimental protocol” again, and then refer to other studies.

7) Around line 96 --> figures or pictures about the experimental setup (wheelchair and sensors) could improve the article.

8) The authors need to better describe the construction of the data set.

9) The metric equations described in lines 136 - 148 could be presented. 

10) Regarding Figure 1. The text in the title needs to abut the figures, but it should not be a paragraph.  Note that in figure B and in the highlight of figure A, the scale should start at “3s and go up to 5.2s”, and not start at zero.

By the way, where are the curves of the Lidar system, as indicated in line 141?

Suggestion, on the x-axis, you can MAINTAIN in seconds, but it would be interesting to have a second (upper) x-axis where the PCs are described / separated.

11) Figure 1 shows the first collection results.... So it's about results, not methodology.

12) line 161: “...using Jamovi software.” --> please, insert a reference here.

13) the equations must be numbered (lines 169 - 170). 

References for these equations are missing. (justify the values, e.g. (1.96 * sqrt(2))

14) “An alpha (α) of 0.05... “ --> Where alpha is used or presented in the paper?

15) The Methodology section need are adjusted.  The entire procedure must be described so that the results presented follow the sequence of the methodology.

For example: the methodology needs to state how the data will be acquired (experimental protocol). After that, how the data will be validated (“convergent validity”) and this is not in the methodology.

Another example: Figure 3... what is the “rank of WRA”? this should have been covered in the methodology... what would be done, why it would be done and how it would be analyzed.

16) line 136 (vs line 188)- “Root mean square difference between Lidar and IMU speed curves (RMSe - Absolute velocity RMS error) were computed for each stride” --> this could appear as an equation and be given a name... as it will be used in figure 2... it is important to make ALL metrics clear and described... so that the results can be better understood and evaluated.

17) In table 1 --> AccGain m/s --> m2/s!

18) line 216 (paragraph indentation) -- there are several points where you need to review the formatting, bold, highlights, subtitles, etc...

19) as already mentioned in item 15 of this review, figure 3 needs to be better described... and here I make an extension to all of them in the article.

20) line 270: “The NP included in our study reached average speed (~ 2.4 m/s)” --> this is not seen in the results presented. It could be highlighted....  (And no results can appear in the discussion that have not been presented before)

21) the dataset needs to be more clear... 

line 19 - “28 studied metrics”

line 275 - “(23/28) of the analyzed metrics”

All metrics should have been clearly indicated in the methodology.

22) The authors MUST separate the discussion from the conclusion. 

==========================

=========================

==========================

=========================

REVIEW 2

=========================

The authors have explicitly introduced what they want to present in the article: “This project aims at developing and evaluating the psychometric properties of spatiotemporal and speed-based metrics using a single IMU positioned on the frame of the wheelchair”

A clear “Methodology” section is still missing, in which the entire procedure must be very well covered.

In this section, it is essential to detail:

a) participants

b) materials 

- with illustrations of the instrumentation of the chairs and the participants, where applicable.

- It is important to say how many wheelchairs were instrumented, whether they were all the same model, whether the instrumentation was the same for all of them.

- it is important to say how the data is collected (equipment, timing, etc.)

c) experimental protocol

- it is important to describe and ILLUSTRATE the experimental protocol, pointing out how the data will be collected, how the actions will be carried out and repeated

d) dataset construction

- it is important to make it clear how the dataset is composed and its size.

e) data processing

- it is important to indicate what will be done with the data, what techniques and/or tools will be used and then insert the statistical analyses that will be evaluated.

(All equations MUST be numbered and properly identified.

Note that here in the methodology, all the analyses in the discussion must be presented and how they will be carried out (e.g. Test-retest reliability, Convergent validity, Comparison with WRA, Limitations, etc).

==========================

Also...the SEM and MDC equations

SEM = Standard deviation of scores / √ ICC  

MDC = SEM x 1.96 (MDC is the smallest change in the score that is statistically significant. It is calculated by multiplying the SEM by 1.96 (which represents the 95% confidence interval).

a) 𝑆𝐸𝑀 = 𝑆𝐷 × √1 - 𝐼𝐶𝐶 (this is different from what we find in the literature ==> 𝑆𝐸𝑀 = 𝑆𝐷 / √ 𝐼𝐶𝐶 

b) (and where does √2 come from ??)

==> you need to explain the components of the equation and mention where they came from.

=========================

=========================

Finally, 

I'd like to say that the authors have done a lot of work to improve some of the previous issues in the article. However, they were really partial to all the revision I sent earlier (the editors can confirm this).  Based on this, I have now emphasized the need to adequately illustrate and describe the methodology, which still falls short of a scientific article.  Another more general issue is still the weak discussion in terms of the content of sensors, their limitations and/or new techniques. I still find the article very tangential to the interests of Sensors readers, and I still don't see a direct (or perhaps, very tangential) correlation with the “Special Issue : Feature Papers in Wearables 2024”. I reiterate that the simple use of sensors and/or statistical applications does not resonate with this journal, and that this topic would be better suited to “Applied Science” or a sports magazine.

Cordially,
